# iFEM2.0: Dense 3D Contact Force Field Reconstruction and Assessment for Vision-based Tactile Sensors

Can Zhao, Jin Liu, and Daolin Ma

*Abstract*— **Vision-based tactile sensors offer rich tactile information through high-resolution tactile images, enabling the reconstruction of dense contact force fields on the sensor surface. However, accurately reconstructing the three-dimensional (3D) contact force distribution remains a challenge. In this paper, we propose the multi-layer inverse finite element method (iFEM2.0) as a robust and generalized approach to reconstruct dense contact force distribution. The proposed iFEM2.0 demonstrates good performance in both simulation- and experiment-based evaluations. Such dense 3D contact force information is critical for enabling dexterous robotic manipulation that handles both rigid and soft materials.**

## I. INTRODUCTION

Robots are increasingly performing complex tasks such as surgical assistance, space servicing, and precision assembly [1]–[3]. In these unstructured environments, dense 3D contact force perception is critical for ensuring safe and effective interaction [4], [5], offering valuable insights into object properties like texture, stiffness, and friction [6]–[9]. While traditional tactile sensors excel at normal force sensing [10]–[12], tangential force measurement is essential for capturing frictional behaviors and detecting slippage [13], [14], particularly when manipulating deformable objects [15], [16]. Fig. 1 illustrates a practical scenario showcasing the importance of dense 3D contact force fields. In this example, the softer outer contact region induces smaller tangential forces on the sensor, whereas the stiffer central part exhibits larger tangential forces, which aligns with Hertzian contact [17] and friction principles [18]. Such nuanced information allows for better perceiving and manipulating rigid-soft coupled objects.

Vision-based tactile sensors [19], [20] offer promising hardware, capable of recovering the surface deformation fields, while accurately estimating the contact force distribution from the deformation fields poses a challenge. This inverse mechanical problem is inherently ill-conditioned, leading to serious errors even with minor measurement noise [21]. Existing methods for reconstructing contact force via vision-based tactile sensors often yield noisy results [22]–[24]. Furthermore, the lack of standardized benchmarks hampers method evaluation and comparison.

This work was jointly supported by the National Natural Science Foundation of China (NSFC) under Grant 12272220, and the Shanghai Jiao Tong University through the Oceanic Interdisciplinary Program under Grant SL2021MS017. *(Corresponding author: Daolin Ma)*

Can Zhao and Daolin Ma are with the School of Ocean & Civil Engineering, Shanghai Jiao Tong University, Shanghai 200240, China. Jin Liu is with the School of Mechanical Engineering, Shanghai Jiao Tong University, Shanghai 200240, China. (e-mail: can.zhxx@sjtu.edu.cn; jinliu.sjtu@outlook.com; daolinma@sjtu.edu.cn)

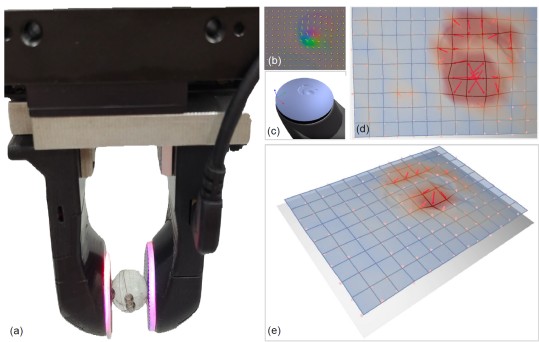

Fig. 1. An example of dense 3D contact force fields for perceiving rigid-soft coupled objects. (a) The GelSlim 3.0 sensor holds onto a coupled rigid-soft ball. (b) 2D tangential displacement field. (c) Depth field. (d) 3D contact force field with noise reconstructed using raw iFEM. (e) Enhanced accuracy and clarity in the 3D contact force field reconstructed via iFEM 2.0.

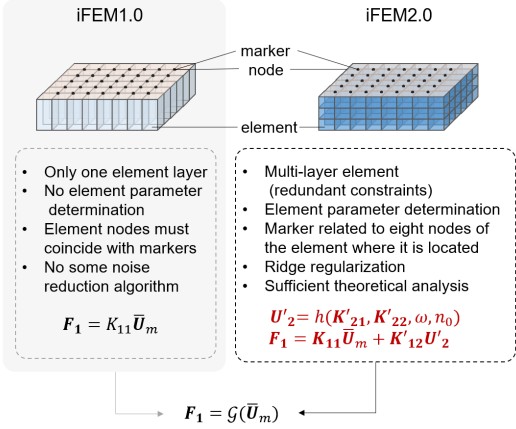

Fig. 2. The iFEM2.0 improvements compared to iFEM1.0. The task is defined as reconstructing the 3D contact force distribution on the pad's surface $\boldsymbol{F}_1$ from the observed 3D deformation field $\boldsymbol{U}_m$.

This work proposes iFEM2.0–a comprehensive and robust method for reconstructing dense contact force fields using vision-based tactile sensors, published in IEEE Transactions on Robotics [25].

## II. METHOD

Fig. 2 visualizes a side-by-side comparison of the iFEM1.0 and iFEM2.0 algorithms, while Fig. 3 presents the overall pipeline of the iFEM2.0 algorithm.

*1) Problem Statement:* The sensor's gel pad is discretized into a multi-layer uniform 8-node hexahedral element system with $n_e$ element layers and $n_p$ node layers ($n_e \geq 2$, $n_p = n_e + 1$). The global displacement and force vectors are denoted as $\boldsymbol{U}$ and $\boldsymbol{F}$, respectively. The observed 3D displacement field on the gel pad surface is $\overline{\boldsymbol{U}}_m$. The goal is

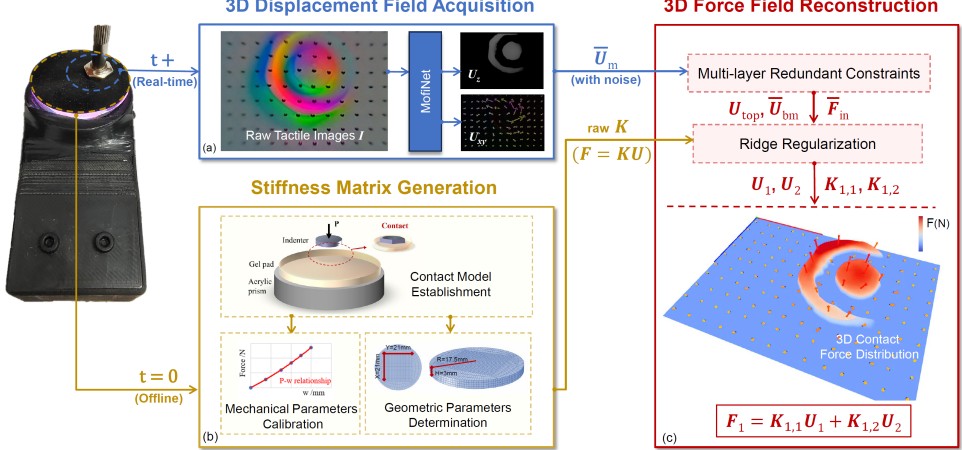

Fig. 3. Algorithm flowchart for 3D contact force distribution. (a) 3D displacement field acquisition from tactile images using a neural network. (b) Stiffness matrix generation: contact model establishment, mechanical parameters calibration, and geometric parameters determination of vision-based tactile sensor. (c) 3D contact force field (normal and tangential) reconstruction from noisy 3D displacement field and original stiffness matrix.

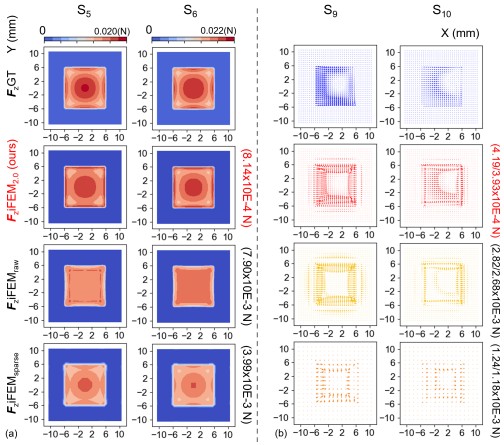

Fig. 4. (a) Normal contour and (b) tangential quiver map for contact force distribution using different methods and states. First row: ground truth, second row: $F_{z\mathrm{iFEM}_{2.0}}$ (our method), third row: $F_{z\mathrm{iFEM}_{\mathrm{raw}}}$, fourth row: $F_{z\mathrm{iFEM}_{\mathrm{sparse}}}$. The columns are labeled with $S_n$, where $n = 5, 6, 9, 10$.

to reconstruct the 3D contact force distribution on the pad's surface $\boldsymbol{F}_1 = \mathcal{G}(\overline{\boldsymbol{U}}_{\mathrm{m}})$, as illustrated in Fig. 2.

*2) Multi-layer Inverse Finite Element Method:* For the only two-layer element case ($n_{\mathrm{e}} = 2$), applying boundary conditions to the finite element equilibrium $\boldsymbol{KU} = \boldsymbol{F}$ yields:

$$\boldsymbol{K}_{2,2}\boldsymbol{U}_2 = -\boldsymbol{K}_{2,1}\boldsymbol{U}_1, \boldsymbol{F}_1 = \boldsymbol{K}_{1,1}\boldsymbol{U}_1 + \boldsymbol{K}_{1,2}\boldsymbol{U}_2. \quad (1)$$

For systems more than two layers ($n_{\mathrm{e}} \geq 3$), intermediate-layer displacements $\boldsymbol{U}_2'$ are condensed, and the corresponding matrices are denoted as $\boldsymbol{K}_{1,2}'$, $\boldsymbol{K}_{2,1}'$, and $\boldsymbol{K}_{2,2}'$. For simplicity, the same notation is used hereafter. However, directly inverting $\boldsymbol{K}_{2,2}$ is ill-conditioned due to its large sparse structure, which amplifies numerical noise and leads to unstable force estimation [21]. In addition, measurement noise in $\overline{\boldsymbol{U}}_{\mathrm{m}}$ further deteriorates reconstruction accuracy.

*3) Ridge Regularization:* To stabilize the inversion, Ridge regularization is introduced for its simplicity, robustness, and numerical efficiency in ill-posed problems [21], [26]. An $L_2$ penalty term is added to the objective, reformulating the force

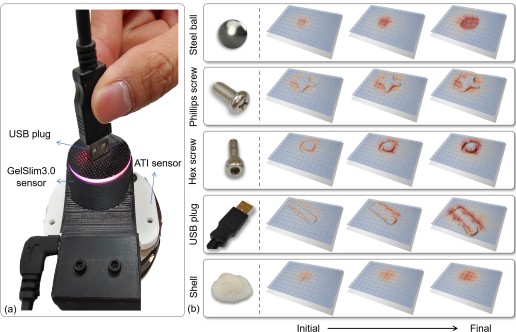

Fig. 5. Practical experimental evaluation and results. (a) Experimental setup with GelSlim 3.0 vision-based tactile sensor and ATI force/torque sensor. (b) Dense 3D contact force distribution reconstructed using iFEM 2.0 for various domestic objects: M8 steel ball, M8 Phillips screw, M6 hexagonal screw, USB plug, and shell.

reconstruction as: $arg\,\min\limits_{\boldsymbol{U}_2}\|\boldsymbol{K}_{2,1}\boldsymbol{U}_1 + \boldsymbol{K}_{2,2}\boldsymbol{U}_2\|_2^2 + \lambda\|\boldsymbol{U}_2\|_2^2$. Solving this convex optimization problem by setting its derivative to zero yields:

$$\boldsymbol{U}_2 = (\boldsymbol{K}_{2,2}{}^{\mathrm{T}}\boldsymbol{K}_{2,2} + w\boldsymbol{I})^{-1}\boldsymbol{K}_{2,2}{}^{\mathrm{T}}(-\boldsymbol{K}_{2,1}\boldsymbol{U}_1 + \boldsymbol{n}_0), \quad (2)$$

where $w$ and $\boldsymbol{n}_0$ denote the regularization parameter and noise level.

## III. CONCLUSION

This study presents iFEM2.0, a multi-layer inverse finite element framework that significantly improves the accuracy and robustness of 3D contact force reconstruction for vision-based tactile sensors. The method enables precise estimation of both normal and tangential force components across dense measurement points. Simulation results in Fig. 4 demonstrate accurate recovery of 3D force fields under varying algorithm settings, while Fig. 5 illustrates high-resolution force distributions captured between the GelSlim3.0 sensor and diverse real-world objects. Beyond reconstruction, iFEM2.0 provides a foundation for tactile-driven closed-loop control, offering new opportunities for adaptive grasping, dexterous manipulation, and complex robotic tasks.

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
