# OpenReview forum: "iFEM2.0: Dense 3-D Contact Force Field Reconstruction and Assessment for Vision-Based Tactile Sensors"
_IEEE.org/IROS/2025/Workshop/Tactile_Sensing — IROS 2025 Workshop Tactile Sensing Poster_

### Official Review · Reviewer_PQYh · 2025-09-13
**Promising method, but lacks results and discussions**

**Rating:** 7
**Confidence:** 5

**Review:**

The authors present a promising force field reconstruction approach using the inverse Finite Element Method (iFEM). While superior performance is claimed, the paper currently lacks results to substantiate this, and the submission would be strengthened by adding a brief section with key outcomes. Additionally, the use of a linear elastic model for the sensor’s gel pad may be a limitation. I encourage the authors to briefly discuss the constraints of this model, particularly for large deformations, and to shed light on future work that could address this.

---

### Official Review · Reviewer_xNiN · 2025-09-25

**Rating:** 7
**Confidence:** 4

**Review:**

This extended abstract proposes iFEM2.0, a pragmatic upgrade to image-based force reconstruction that models the gel with multilayer 8-node hexahedral meshes and stabilizes the ill-posed inverse problem via ridge regularization, paired with in-situ calibration and a four-part benchmark (accuracy, fidelity, noise robustness, generalizability). Strengths: the problem framing is clear; the pipeline is well engineered and likely more stable than single-layer iFEM; and the emphasis on practical calibration/benchmarking improves real-world usability. Limitations: the novelty over prior iFEM is incremental and not yet isolated by theory or ablations; boundary physics and constraints (e.g., friction cone, negative normal forces, edge artifacts) are under-specified; and key evaluation details (ground truth acquisition, statistical significance, parameter sensitivity) are missing, leaving generalization claims insufficiently substantiated.